# Reliable Causal Discovery with Improved Exact Search and Weaker Assumptions

**Ignavier Ng**[1], **Yujia Zheng**[1], **Jiji Zhang**[2], **Kun Zhang**[1]
[1] Carnegie Mellon University
[2] Hong Kong Baptist University
{ignavierng, yujiazh}@cmu.edu, zhangjiji@hkbu.edu.hk, kunz1@cmu.edu

## Abstract

Many of the causal discovery methods rely on the faithfulness assumption to guarantee asymptotic correctness. However, the assumption can be approximately violated in many ways, leading to sub-optimal solutions. Although there is a line of research in Bayesian network structure learning that focuses on weakening the assumption, such as exact search methods with well-defined score functions, they do not scale well to large graphs. In this work, we introduce several strategies to improve the scalability of exact score-based methods in the linear Gaussian setting. In particular, we develop a super-structure estimation method based on the support of inverse covariance matrix which requires assumptions that are strictly weaker than faithfulness, and apply it to restrict the search space of exact search. We also propose a local search strategy that performs exact search on the local clusters formed by each variable and its neighbors within two hops in the super-structure. Numerical experiments validate the efficacy of the proposed procedure, and demonstrate that it scales up to hundreds of nodes with a high accuracy.

## 1   Introduction

Although it is often more reliable to discover causal relationships by making use of interventions or randomized experiments, they are practically challenging, expensive, or even prohibited owing to ethical considerations. Thus, causal discovery from observational data has received considerable attention in recent decades, and has been widely applied in different fields such as genetics [29].

One major class of causal discovery methods is the constraint-based methods, such as PC [40] and FCI [43, 4], that leverage conditional independence tests to estimate the skeleton and then perform edge orientation. These methods are guaranteed to asymptotically return the true Markov equivalence class (MEC) under the Markov and faithfulness assumptions. Several modifications [30, 41] to these constraint-based methods have been developed to allow certain types of unfaithfulness, which, however, generally give rise to weaker claims and are not guaranteed to estimate the true MEC.

Another popular approach is the GES [3] algorithm that searches in the space of MECs greedily by maximizing a well-defined score, such as the Bayesian information criterion (BIC) [36] score. It starts with an empty structure and consists of two phases: (1) adding edges until a local maximum is found, and (2) removing edges until a local maximum is reached. In spite of the greedy strategy, GES converges in the large sample limit to the true MEC under the Markov and faithfulness assumptions, similar to the aforementioned constraint-based methods.

Recently, NOTEARS [52] casts the Bayesian network structure learning task into a continuous constrained optimization problem with the least squares objective, using an algebraic characterization of directed acyclic graph (DAG). Subsequent work GOLEM [23] adopts a continuous unconstrained optimization formulation with a likelihood-based objective. For NOTEARS, it remains unclear

35th Conference on Neural Information Processing Systems (NeurIPS 2021).

of the required assumptions for asymptotic correctness, whereas GOLEM adopts the generalized faithfulness assumption [9] to learn linear Gaussian DAGs, which could be converted to their MECs for causal interpretation [42]. These methods enable the application of numerical solvers and GPU acceleration, which thus are scalable to large graphs. However, they are only guaranteed to find a local optimum of the optimization problem, and therefore the quality of the solution in practice may not be guaranteed, even in the asymptotic case.

Another line of research focuses on weakening the faithfulness assumption required for asymptotic correctness of the search results, since, given finite samples, approximate violations of faithfulness occur surprisingly often, especially when there is a large number of variables [46]. For instance, exact search methods find the optimal Bayesian network based on a predefined score function, such as dynamic programming (DP) [18, 24, 39, 38], A* [49, 48], and integer programming [1, 5]. The DAGs estimated by these methods can be converted to their MECs for causal interpretation [42]. Note that the approaches based on sparsest permutation (SP) [32] and Boolean satisfiability solver (SAT) [15, 16] can be viewed as instances of exact methods. Lu et al. [22] further demonstrated that these exact methods may produce correct results in cases where methods relying on faithfulness fail.

Due to the large search space of possible DAGs [2, 13], exact search methods are feasible only for small structures. Therefore, *super-structure* has been adopted to constrain the search space [27, 45], which is defined to be an undirected graph that restricts the search to candidate DAGs whose skeleton is its subgraph. However, most of these methods rely on discovering the skeleton of the true DAG for use as a super-structure, utilizing estimation methods like MMPC [45], which require the faithfulness assumption to be asymptotically correct. Under approximate violations of faithfulness, these skeleton estimation methods may miss some edges owing to unfaithful conditional independencies in the data distribution; thus, further exact search procedures are guaranteed to miss those edges.

**Contributions.** In this work, we introduce several strategies to improve the scalability of exact search in the linear Gaussian setting, giving rise to a more reliable causal discovery procedure. Our main contributions can be summarized as follows:

- We develop a super-structure estimation method based on the support of inverse covariance matrix of the data distribution, and show that it is asymptotically correct under assumptions strictly weaker than faithfulness (or, more specifically, than triangle-faithfulness). We combine this with exact search method like DP or A* to reduce search space.

- To further scale up exact search, we develop a local search strategy, called Local A*, on the local clusters formed by each variable and its neighbors within two hops in the super-structure.

- We demonstrate the efficacy of our super-structure estimation method and local search strategy by conducting extensive experiments, and show that it scales up to hundreds of nodes with a high accuracy.

**Paper organization.** We review the common assumptions for causal discovery and the linear structural equation model (SEM) in Section 2. In Section 3, we establish weaker variants of faithfulness and show how they could be used to learn a sound super-structure. We further formulate an improved exact search strategy in Section 4. The empirical studies in Section 5 validate our theoretical results and the efficacy of the proposed procedure. We then conclude our work in Section 6.

## 2 Background

We first review the concepts of causal Bayesian networks and some commonly used assumptions that are related to our further analysis. We then give a brief overview of the linear SEM.

### 2.1 Causal Bayesian Network and Common Assumptions

Let $\mathcal{G} = (\mathbf{V}, \mathbf{E})$ be a DAG with vertex set $\mathbf{V} = \{X_1, \dots, X_d\}$ in which each node $X_i$ corresponds to a random variable. Denote $\mathbf{X} = (X_1, \dots, X_d)$ as the random vector concatenating all variables and its associated probability distribution $\mathbb{P}$. Let $\mathbf{X}_{pa(i)}$ be the set of parental nodes of $X_i$ in $\mathcal{G}$ such that there is a directed edge from $X_j \in \mathbf{X}_{pa(i)}$ to $X_i$, or $X_j \rightarrow X_i \in \mathbf{E}$. We assume *causal sufficiency*, i.e., no hidden variables, throughout the paper.

In a *Bayesian network*, the distribution $\mathbb{P}$ is assumed to be Markov w.r.t. to DAG $\mathcal{G}$, as defined below.

**Assumption 1 (Markov).** *Given a DAG $\mathcal{G}$ and distribution $\mathbb{P}$ over the variable set $\mathbf{V}$, every variable $X$ in $\mathbf{V}$ is probabilistically independent of its non-descendants given its parents in $\mathcal{G}$.*

A *causal Bayesian network* can be viewed as a Bayesian network where the directed edges are provided a causal meaning, which thereby allows it to answer interventional queries [20]. In general, there are many DAGs that induce the same conditional independence (CI) relations with the distribution $\mathbb{P}$, and are said to be *Markov equivalent*. The *Markov equivalence class* (MEC) consists of all DAGs that entail the same conditional independence (CI) relations as $\mathcal{G}$ does, and is uniquely determined by its skeleton and v-structures [25]. *V-structure* is defined to be a collider $X \rightarrow Y \leftarrow Z$ where $X$ and $Z$ are not adjacent in $\mathcal{G}$, therefore referred to as an *unshielded collider*. If $X$ and $Z$ are adjacent, then it is called a *shielded collider*.

The following faithfulness assumption is commonly used to relate the CI relations in the distribution to the DAG, and can be thought of as the converse to the Markov assumption.

**Assumption 2 (Faithfulness [44]).** *Given a DAG $\mathcal{G}$ and distribution $\mathbb{P}$ over the variable set $\mathbf{V}$, $\mathbb{P}$ implies no CI relations not already entailed by the Markov assumption.*

Under the Markov and faithfulness assumptions, constraint-based methods such as PC have been shown to asymptotically output the correct MEC. However, in the finite sample regime, the faithfulness assumption is sensitive to statistical testing errors when inferring the CI relations, and its approximate violations occur surprisingly often, especially when there is a large number of variables [46]. Thus, different relaxations of faithfulness have been proposed, such as orientation-faithfulness, adjacency-faithfulness [30], and triangle-faithfulness [51], which we review in Appendix A.

Clearly, the triangle-faithfulness assumption is a consequence of adjacency-faithfulness. Based on these weaker assumptions, different modifications to constraint-based methods have been proposed, such as Conservative PC [30] and Very Conservative SGS [41]. As their names suggest, these methods make weaker claims about the estimated causal structure and therefore are not guaranteed to estimate the true MEC. On the other hand, Raskutti and Uhler [32] proposed the following assumption and show that it is strictly weaker than faithfulness.

**Assumption 3 (Sparsest Markov representation (SMR) [32]).** *Given a DAG $\mathcal{G}$ and distribution $\mathbb{P}$ over the variable set $\mathbf{V}$, the MEC of $\mathcal{G}$ is the unique sparsest MEC that satisfies the Markov assumption with $\mathbb{P}$.*

Forster et al. [7] referred to the above assumption as the (unique) frugality assumption, and argued that it has multiple desirable properties compared to faithfulness. Under the SMR assumption, the SP method [32] has been shown to produce asymptotically correct results. Raskutti and Uhler [32] also conjectured that SP reaches the information-theoretic limit in the sense that the SMR assumption may be the weakest assumption guaranteeing the asymptotic correctness of any method for learning the true MEC. It is worth noting that SP can be viewed as an instance of exact score-based method. The study by Lu et al. [22] demonstrated that causal discovery methods that rely on the faithfulness assumption (e.g., PC, GES) may output sub-optimal solutions in various cases, whereas exact methods (e.g., SP, A*, SAT) are able to produce the correct results. Therefore, in this work, we aim at improving the scalability of exact score-based methods for reliable causal discovery, by relying on the SMR assumption and relaxing the faithfulness assumption as much as is viable.

## 2.2 Linear Structural Equation Model

Given a linear SEM, each random variable follows the relationship $X_i = \mathbf{B}_i^\top \mathbf{X} + N_i$, where $\mathbf{B}_i$ is a coefficient vector and $N_i$ is an exogenous noise variable associated with variable $X_i$. In this work we focus on the linear Gaussian model where the variables $N_i$'s follow the Gaussian distribution. The linear SEM can be written in matrix form as $\mathbf{X} = \mathbf{B}^\top \mathbf{X} + \mathbf{N}$, where $\mathbf{B} = [\mathbf{B}_1, \mathbf{B}_2, \cdots, \mathbf{B}_d]$ corresponds to a weighted adjacency matrix, and $\mathbf{N} = (N_1, \ldots, N_d)$ is a noise vector characterized by the covariance matrix $\mathbf{\Omega} = \text{cov}[\mathbf{N}] = \text{diag}(\sigma_1^2, \ldots, \sigma_d^2)$. We assume that $\sigma_i^2 > 0$ for $i = 1, \ldots, d$ so that the distribution $\mathbb{P}$ has positive measure everywhere. As a standard assumption, we also assume *structural minimality* [29] which implies that the nonzero coefficients in $\mathbf{B}$ define the structure of $\mathcal{G}$, i.e., $X_j \rightarrow X_i \in \mathbf{E}$ if and only if the coefficient in $\mathbf{B}_i$ corresponding to variable $X_j$ is nonzero. Since one can always center the data, we assume $\text{E}[\mathbf{X}] = \text{E}[\mathbf{N}] = 0$ without loss of generality. The inverse covariance matrix $\mathbf{\Theta} = \mathbf{\Sigma}^{-1}$ of $\mathbf{X}$ is given by $\mathbf{\Theta} = (\mathbf{I} - \mathbf{B})\mathbf{\Omega}^{-1}(\mathbf{I} - \mathbf{B})^\top$. Note that $\mathbf{\Theta}_{ij} = 0$ if and only if $X_i \perp\!\!\!\perp X_j | \mathbf{V} \setminus \{X_i, X_j\}$ in the linear Gaussian case.

# 3 Weaker Assumptions for Super-Structure Estimation

We establish several weaker variants of faithfulness assumption, and show how they are both necessary and sufficient for learning a super-structure of the true DAG via inverse covariance estimation.

## 3.1 Weaker Assumptions than Faithfulness

We describe several relaxed assumptions of faithfulness required for our super-structure estimation procedure. We first start with the specific types of faithfulness related to different kinds of colliders.

**Assumption 4 (Shielded-collider-faithfulness (SCF)).** *Given a DAG $\mathcal{G}$ and distribution $\mathbb{P}$ over the variable set $\mathbf{V}$, let $X \to Y \leftarrow Z$ be any shielded collider in $\mathcal{G}$. Then $X$ and $Z$ are dependent conditional on any subset of $\mathbf{V} \setminus \{X, Z\}$ that contains $Y$.*

**Assumption 5 (Unshielded-collider-faithfulness (UCF)).** *Given a DAG $\mathcal{G}$ and distribution $\mathbb{P}$ over the variable set $\mathbf{V}$, let $X \to Y \leftarrow Z$ be any unshielded collider in $\mathcal{G}$. Then $X$ and $Z$ are dependent conditional on any subset of $\mathbf{V} \setminus \{X, Z\}$ that contains $Y$.*

The above assumptions are restrictions of the triangle-faithfulness [51] and orientation-faithfulness [30] assumptions, respectively, to collider structures. Note that they differ only in the type of collider being considered. However, these assumptions require dependence conditioning on any subset of $\mathbf{V} \setminus \{X, Z\}$, which may be restrictive in practice. In this work, we further relax them to require only dependence conditioning on $\mathbf{V} \setminus \{X, Z\}$, and will show in Section 3.2 how they are both necessary and sufficient conditions for estimating a sound super-structure of the true DAG. These relaxed assumptions are stated below.

**Assumption 6 (Single shielded-collider-faithfulness (SSCF)).** *Given a DAG $\mathcal{G}$ and distribution $\mathbb{P}$ over the variable set $\mathbf{V}$, let $X \to Y \leftarrow Z$ be any shielded collider in $\mathcal{G}$. Then $X \not\perp\!\!\!\perp Z | \mathbf{V} \setminus \{X, Z\}$.*

**Assumption 7 (Single unshielded-collider-faithfulness (SUCF)).** *Given a DAG $\mathcal{G}$ and distribution $\mathbb{P}$ over the variable set $\mathbf{V}$, let $X \to Y \leftarrow Z$ be any unshielded collider in $\mathcal{G}$. Then $X \not\perp\!\!\!\perp Z | \mathbf{V} \setminus \{X, Z\}$.*

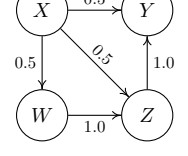

Figure 1 illustrate the examples in which the SCF and UCF assumptions are violated, respectively, but the SSCF and SUCF assumptions hold. In these examples, we have $X \perp\!\!\!\perp Z | Y$ and $X \not\perp\!\!\!\perp Z | \{Y, W\}$, where the former unfaithful CI relation $X \perp\!\!\!\perp Z | Y$ is constructed via path cancellations.

(a) Violation of SCF.

Given a distribution $\mathbb{P}$, let $\mathcal{G}_{fai}(\mathbb{P})$, $\mathcal{G}_{adj}(\mathbb{P})$, $\mathcal{G}_{ori}(\mathbb{P})$, $\mathcal{G}_{tri}(\mathbb{P})$, $\mathcal{G}_{scf}(\mathbb{P})$, $\mathcal{G}_{ucf}(\mathbb{P})$, $\mathcal{G}_{sscf}(\mathbb{P})$, and $\mathcal{G}_{sucf}(\mathbb{P})$ be the set of DAGs satisfying the faithfulness, adjacency-faithfulness, orientation-faithfulness, triangle-faithfulness, SCF, UCF, SSCF, and SUCF assumptions, respectively. Also, denote by $\subset$ the proper subset symbol. Combining with the results by Ramsey et al. [30], Zhang and Spirtes [51], we have the following nesting properties, showing that SSCF and SUCF are intuitively much weaker than the faithfulness assumption.

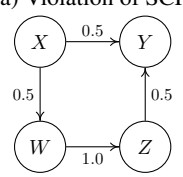

**Remark 1.** $\mathcal{G}_{fai}(\mathbb{P}) \subset \mathcal{G}_{adj}(\mathbb{P}) \subset \mathcal{G}_{tri}(\mathbb{P}) \subset \mathcal{G}_{scf}(\mathbb{P}) \subset \mathcal{G}_{sscf}(\mathbb{P})$.

**Remark 2.** $\mathcal{G}_{fai}(\mathbb{P}) \subset \mathcal{G}_{ori}(\mathbb{P}) \subset \mathcal{G}_{ucf}(\mathbb{P}) \subset \mathcal{G}_{sucf}(\mathbb{P})$.

(b) Violation of UCF.

Figure 1: Examples.

## 3.2 Inverse Covariance Estimation for Learning Super-Structure

Based on the assumptions described in Section 3.1, we develop a super-structure estimation method via inverse covariance estimation. We first study the specific assumptions required for the support of inverse covariance, denoted as $\text{supp}(\mathbf{\Theta})$, to be the same as the moralized graph of the underlying DAG. This is similar to the analysis by Loh and Bühlmann [21], but we focus on formulating precisely the required assumptions from a causal discovery perspective, to shed light on how weak they are as compared to faithfulness. This enables us to further weaken the required assumptions in order to recover a super-structure of the true DAG based on the support of inverse covariance.

As described, we first have the following theorem that relates the moral graph and the support of inverse covariance to the SUCF and SSCF assumptions, with a proof given in Appendix C.1.

**Theorem 1.** *Given a DAG $\mathcal{G}$ and distribution $\mathbb{P}$ that follow a linear Gaussian model with inverse covariance matrix $\mathbf{\Theta}$, under Markov assumption, the SSCF and SUCF assumptions are satisfied if and only if the structure defined by $\text{supp}(\mathbf{\Theta})$ is the same as the moralized graph of the true DAG $\mathcal{G}$.*

Although the above theorem guarantees recovering the moral-ized graph via $\mathrm{supp}(\boldsymbol{\Theta})$, in practice the SUCF assumption may be approximately violated. To illustrate, we simulate the lin-ear Gaussian model with edge weights sampled uniformly from $[-0.8, -0.2] \cup [0.2, 0.8]$ and exogenous noise variances sampled uniformly from $[1, 2]$. We then compute the minimum values $\min_{i,j}\{|\boldsymbol{\Theta}_{ij}| : X_i$ and $X_j$ correspond to a pair of neighbors in $\mathcal{G}\}$ and $\min_{i,j}\{|\boldsymbol{\Theta}_{ij}| : X_i$ and $X_j$ correspond to a pair of non-adjacent spouses in $\mathcal{G}\}$ over 100 simulations, by considering different expected degrees and number of variables $d \in$

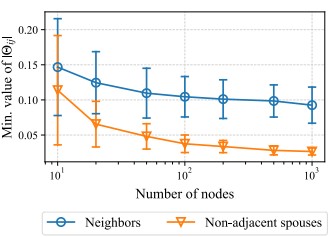

Figure 2: Expected degree of 2.

$\{10, 20, 50, 100, 200, 500, 1000\}$. Note that $X_i$ and $X_j$ being a pair of non-adjacent spouses in DAG $\mathcal{G}$ implies that they share a common child and have an unshielded collider; therefore, SUCF requires that $X_i \not\perp\!\!\!\perp X_j | \mathbf{V} \setminus \{X_i, X_j\}$, i.e., $\boldsymbol{\Theta}_{ij} \neq 0$ in the linear Gaussian case. The visualizations for expected degree of 2 is shown in Figure 2, while those for degrees of 5 and 8 can be be found in Appendix B. Although the values in both cases decrease with larger graphs,[1] the minimum values corresponding to the non-adjacent spouses are significantly smaller than those of neighbors, where the difference grows in the number of nodes. For instance, on 1000-node graphs with degree of 2, the average minimum value of the former is $0.025$, while that of the latter is close to $0.1$. With finite samples, it is very challenging to discover those undirected edges in the former case, since the weights are very close to zero, leading to approximate violations of SUCF especially when there is a large number of variables. Indeed, the simulations above are based on a certain range of edge weights and noise variances, whose true values are not known in practice. Therefore, it is possible that the data distribution considered falls into similar setting, leading to violation of SUCF. It is thus desirable to further weaken the required assumptions to develop a more reliable procedure.

If we drop the SUCF assumption, the structure defined by $\mathrm{supp}(\boldsymbol{\Theta})$ may miss some edges as compared to the moralized graph of the true DAG. Fortunately, these edges correspond to the non-adjacent spouses in the DAG. That is, $\mathrm{supp}(\boldsymbol{\Theta})$ is still guaranteed to contain all undirected edges corresponding to neighbors in the DAG, as long as SSCF (in addition to the Markov assumption) holds, indicating that it is a sound super-structure of the true DAG. Based on the simulations considered in Figure 2, approximate violations of SSCF are much less likely to happen than those of SUCF.

**Theorem 2.** *Given a DAG $\mathcal{G}$ and distribution $\mathbb{P}$ that follow a linear Gaussian model with inverse covariance matrix $\boldsymbol{\Theta}$, under Markov assumption, the SSCF assumption is satisfied if and only if the structure defined by $\mathrm{supp}(\boldsymbol{\Theta})$ is a super-structure of the true DAG $\mathcal{G}$.*

The proof is provided in Appendix C.1. With an asymptotically correct approach to estimate $\mathrm{supp}(\boldsymbol{\Theta})$, the theorem implies that one can estimate a sound super-structure of the true DAG under the SSCF assumption that is strictly weaker than triangle-faithfulness (cf. Remark 1), an assumption which is intuitively much weaker than faithfulness. This may be considered as a significant improvement over the existing methods that estimate the skeleton of the true DAG for use as a super-structure, utilizing methods like MMPC [45], which require faithfulness to be asymptotically correct.

The question remains as how to estimate $\mathrm{supp}(\boldsymbol{\Theta})$. In the high-dimensional setting, the $\ell_1$-penalized maximum likelihood estimator can be used to recover the support, with consistency result provided by Ravikumar et al. [33, 34]. In this work, we adopt the widely used graphical Lasso (GLasso) [8] method based on the block coordinate descent algorithm for estimating $\mathrm{supp}(\boldsymbol{\Theta})$. Other efficient estimation method, e.g., QUIC [14], can also be adopted, which is treated as a future work.

We provide empirical studies in Appendix E.1, in which faithfulness is *exactly* violated but not SSCF, to demonstrate that GLasso finds the true super-structure in these cases, whereas MMPC fails. This is consistent with the experiments in Section 5.1 which show that GLasso is more reliable in practice, since *approximate* violations of SSCF are less likely to happen as compared to those of faithfulness.

## 4 Improved Exact Search

We introduce several strategies to improve the scalability of exact score-based search. In Section 4.1, we first establish the connection between the SMR assumption and exact search with BIC. We then leverage the super-structure estimation method described in Section 3.2 to restrict the search space of exact methods in Section 4.2, and develop a local search strategy in Section 4.3.

---

[1]Note that this is consistent with the analysis of the strong faithfulness assumption by Uhler et al. [46].

### 4.1 Exact Search and the SMR Assumption

As described in Section 2.1, classical methods such as PC and GES may produce sub-optimal solutions when faithfulness fails. In the following, we show the connection between the SMR assumption, which is strictly weaker than faithfulness, and the asymptotic correctness of exact search with BIC. To the best of our knowledge, this is the first time that this result has been formally established.

**Theorem 3.** *Exact score-based search with BIC asymptotically outputs a DAG that belongs to the MEC of the true DAG $\mathcal{G}$ if and only if the DAG $\mathcal{G}$ and distribution $\mathbb{P}$ satisfy the SMR assumption.*

The proof can be found in Appendix C.2, which is straightforward from the consistency of BIC [12, 3]. Under the SMR assumption, the DAG estimated by exact search with BIC, is not necessarily identical to, but belongs to the same MEC as, the true DAG $\mathcal{G}$. Therefore, one has to convert the estimated DAG to its MEC for causal interpretation [42].

### 4.2 A* with Super-Structure

Although exact search methods rely on the SMR assumption that is strictly weaker than faithfulness, it is challenging to scale them up to large graphs as the task is NP-hard [2]. To remedy this issue, similar to [27], we propose to explicitly constrain the search space of exact score-based search algorithms using super-structure. This is achieved by limiting the size of parent graphs in the search procedure.

To give a brief overview, for exact score-based methods such as DP [39, 38] and A* [49, 48], *parent graph* plays an important role in constructing the search space, which is a data structure that stores the costs for the arcs of the order graph. These methods typically construct a parent graph for each variable, which is used to build the *order graph*. The search is performed on the order graph [39, 48], by solving a shortest path problem from the root node, which corresponds to the empty set, to the leaf node, which corresponds to the complete set of variables. Each variable has its own parent graph, and all optimal parent sets are selected from the candidate parent sets, by selecting the ones with the best score in the corresponding parent graph. Here we omit the details of the parent graph and order graph, and refer the interested reader to the references above for their complete definitions and procedure.

Using the super-structure estimation method in Section 3.2, for each variable, we are able to obtain the set of nodes that must not be its candidate parents, and directly remove the entries involving these nodes in the corresponding parent graph. These constrained parent graphs help reduce search space in the order graph, from which the shortest path problem is formulated, and thereby improve the efficiency of the exact search procedure. As a byproduct, this strategy also reduces the memory cost, since one does not have to enumerate all candidate parent sets when constructing the parent graphs.

Note that DP and A* differ mainly in the search phase of the order graph. That is, DP has to consider all combinations of nodes (i.e., $2^d$ combinations), and therefore is exponentially expensive in computation and quickly becomes infeasible when the graph size increases, while A* uses a heuristic function to only expand the most promising node [48], and has been shown empirically to achieve better efficiency. Therefore, in this work we adopt the A* algorithm with the incorporated super-structure to further constrain the search space, which we call *A\*-SS*, although similar strategy also works for DP. Under the Markov and SSCF assumptions, our estimated super-structure via inverse covariance estimation is guaranteed to asymptotically contain the skeleton of the ground-truth DAG (cf. Theorem 2); thus, the proposed constrained search will not suffer from the trade-off between scalability and reliability given a sufficient number of samples.

Besides using super-structure, we apply several existing strategies to improve the efficiency and memory cost of A*. In particular, we use the sparse representations [48] of the parent graphs to further remove unnecessary entries, which have been shown to improve the efficiency and substantially reduce the memory cost. We also adopt the optimal path extension [17] and dynamic k-cycle conflict heuristics [48] to reduce search space during the A* search. We refer the interested reader to the aforementioned references for detailed explanations of these strategies.

### 4.3 Local A*

Even with super-structure and the other techniques described in Section 4.2, scaling up exact score-based search remains a challenge because of the large search space. Given the popularity of distributed computation, the idea of divide-and-conquer has been explored for scaling up Bayesian network

structure learning [10, 50, 19, 47]. However, most of them involve conditional independence tests, which therefore rely on the faithfulness assumption in some way.

To avoid relying on the faithfulness assumption, we aim to develop a local search strategy based on exact search that relies on the SMR assumption. Lu et al. [22] proposed an approximation algorithm, called Triplet A*, to do so. In particular, they first apply estimation method like MMPC [45] to obtain a skeleton. For each variable $X$ and each pair of its neighbors $Y$ and $Z$, the algorithm constructs a cluster that consists of variables $X, Y, Z$ and their neighbors indicated by the skeleton, and runs exact score-based search such as A* on each of these clusters independently. Finally, the final structure is obtained by combining the search results from all clusters. A possible drawback is that the skeleton estimation method like MMPC usually requires the faithfulness assumption, although Triplet A* has been shown to be able to recover some of the missing edges caused by the unfaithful CI relations.

Without faithfulness, fortunately, one is still able to obtain a super-structure of the underlying DAG using our procedure under a strictly weaker assumption. In particular, the proposed super-structure estimation method involving inverse covariance estimation described in Section 3.2 requires only the SSCF assumption that is intuitively much weaker than faithfulness. Therefore, our goal is to develop a local search strategy based on this super-structure estimation method.

Following the similar strategy in [22], our main idea is that, for any variable $X$, its parents, children, spouses, and grandparents contain sufficient information for exact score-based search (with SMR assumption) to correctly discover the undirected edges and v-structures involving $X$ (if there is any). This is because this variable set, denoted as $\mathbf{V}_X$, includes all variables that are direct common causes of $X$ and any of its direct neighbors. The question is then how to correctly identify a set that contains those variables in $\mathbf{V}_X$. A naive approach is to apply skeleton estimation methods like MMPC [45] to estimate a skeleton of the true DAG $\mathcal{G}$. Clearly, for each variable $X$, its neighbors within two hops in the skeleton are guaranteed to contain its parents, children, spouses, and grandparents. However, MMPC requires the faithfulness assumption to be asymptotically correct, which may be restrictive in practice. Fortunately, based on Theorem 2, under the Markov and SSCF assumptions, the structure defined by $\mathrm{supp}(\mathbf{\Theta})$, denoted as $\mathcal{G}(\mathrm{supp}(\mathbf{\Theta}))$, is asymptotically a super-structure of the true DAG. It straightforwardly follows that the neighbors of variable $X$ within two hops in $\mathcal{G}(\mathrm{supp}(\mathbf{\Theta}))$ must contain its parents, children, spouses, and grandparents. These variables, together with $X$ itself, then contain sufficient information for exact score-based search to correctly discover the undirected edges and v-structures involving $X$. The empirical studies in Section 5.2 suggest that the proposed local search procedure is asymptotically correct as it returns the same solutions as the A*-SS method.

As described in Section 3.2, in practice, one can use the GLasso method to produce an estimate of the inverse covariance matrix $\widehat{\mathbf{\Theta}}$, and obtain its defined structure, denoted as $\mathcal{G}(\mathrm{supp}(\widehat{\mathbf{\Theta}}))$. Then, we construct a *local cluster* for each variable, which consists of the variable itself and its neighbors within two hops in the structure $\mathcal{G}(\mathrm{supp}(\widehat{\mathbf{\Theta}}))$, and apply an exact score-based method such as A* on these local clusters independently. We then obtain the final structure by combining these local results.

Our approach can be easily parallelized by running the local searches for all variables in *parallel*, in which the complexity depends on the maximum size of local clusters. In practice, the computational resources may be limited and one is not able to do so. Furthermore, there is a huge overlap between different local clusters, leading to the redundancy of computation during the search procedure. To alleviate this, we consider an *iterative* approach starting from the smallest local cluster to the largest one that keeps some edges fixed to take full advantage of the information obtained from the previous searches. More specifically, in each iteration, we save the discovered undirected edges and v-structures involving the target variable from the local search. Then in the next iteration for, e.g., variable $X$, we look up the undirected edges and v-structures involving $X$ that have been discovered and saved from previous iterations, and keep these edges fixed in the local search for variable $X$. By trusting the information conveyed by these edges and keeping them fixed, the search space for the following clusters can be drastically constrained. This improved procedure, called *Local A*, is illustrated in Algorithm 1. Note that one could run some of the local searches in parallel to accelerate the iterative procedure, depending on the available computational resources.

It may be challenging to derive the exact computational complexity of the proposed method, as is the case for the A* algorithm, partly owing to the heuristic involved [11, Section 4]. This is because the complexity is affected by different factors such as the size of sparse parent graphs and the heuristic function (i.e., dynamic k-cycle conflict heuristics) [48]. Thus, we are only able to provide the running time as a proxy of the computational complexity in Section 5.4.

---

**Algorithm 1** Local A*

---

**Require:** Structure defined by the support of estimated inverse covariance matrix $\mathcal{G}(\text{supp}(\widehat{\mathbf{\Theta}}))$.

1: Initialize $\widehat{\mathbf{M}}$ as a $d \times d$ matrix in which all entries are zero.
2: Let $N(i)$ denote the set of neighbors of variable $X_i$ in the structure $\mathcal{G}(\text{supp}(\widehat{\mathbf{\Theta}}))$.
3: Obtain a variable ordering $\pi$ by sorting the variables $X_i, i = 1, \ldots, d$ based on the cardinality of their corresponding local cluster $C_i = \{X_i\} \cup N(i) \cup \left( \bigcup_{j \in N(i)} N(j) \right)$. Specifically, $\pi(j)$ refers to the index such that $C_{\pi(j)}$ is the $j$-th smallest local cluster.
4: **for** $i = \pi(1), \pi(2), \ldots, \pi(d)$ **do**
5:     Obtain the local cluster $C_i$.
6:     From $\widehat{\mathbf{M}}$, obtain the previously discovered undirected edges and v-structures involving $X_i$.
7:     Orient the undirected edges such that they do not create any cycle or additional v-structure.
8:     With these edges fixed, run A* (or A*-SS) on cluster $C_i$.
9:     Convert the DAG estimated by A* to its MEC, and save the newly discovered undirected
10:       edges and v-structures involving variable $X_i$ to $\widehat{\mathbf{M}}$.
11: **end for**
12: Output the matrix $\widehat{\mathbf{M}}$ that represents the final MEC.

---

## 5 Experiments

We first conduct experiments to compare the efficacy of GLasso and MMPC for estimating super-structures of the true DAGs. We then validate the proposed search strategies, and compare them to different baselines. Lastly, we compare different approaches to scale up A* search, including the proposed ones. The baselines include Triplet A* [22], PC [40, 30], FGES [31], MMHC [45], and SP [32]. The implementation details of our method and the baselines are given in Appendix D.

In our experiments, the ground truth DAGs are simulated using the Erdös–Rényi model [6] with different degrees and number of variables. We construct the weighted adjacency matrix of each DAG using edge weights sampled uniformly from $[-0.8, -0.2] \cup [0.2, 0.8]$. Based on the weighted matrix constructed, we simulate $n \in \{300, 10000\}$ samples using the linear Gaussian model with exogenous noise variances sampled uniformly from $[1, 2]$. We report the structural Hamming distance (SHD) over the complete partial DAGs (CPDAGs). We also compute the F1 score of the undirected and directed edges in the estimated CPDAGs. We do not provide the complete results for structural intervention distance (SID) [28], and include only a summary in Appendix E.2. Unless otherwise stated, we report the results and standard errors computed over 10 random simulations.

### 5.1 Different Super-Structure Estimation Methods

To demonstrate the efficacy of GLasso for estimating super-structures with weaker assumptions, we compare the quality of super-structures estimated by GLasso to those by MMPC. We evaluate their ability for discovering the *direct neighbors* in the ground truth, by computing the true positive rates (TPRs) and false discovery rates (FDRs) of estimating the true neighbors. Although the output of GLasso may contain non-adjacent spouses, here we only evaluate its discovered direct neighbors as only those are important for further exact search. We consider 10-node graphs with varying sample sizes and expected degrees, and report the results of MMPC with different significance levels $\alpha$.

Due to space limit, the results for sample size $n = 300$ are shown in Figure 3, while those for $n = 10000$ are reported in Figure 7 in Appendix E.3. In the first panel of Figure 3, one observes that GLasso achieves TPRs close to one across all cases, indicating that it rarely misses any direct neighbor, as compared to MMPC with lower TPRs. Notice also that the difference grows as the degree increases. Although GLasso has slightly higher FDRs than MMPC, we believe that the higher TPRs justify the cost of having a larger search space for exact search. It is interesting to observe that even with a high significance level $\alpha$ such as $0.5$ (i.e., the hypothesis tests used by MMPC tend to produce more edges), MMPC has low TPRs on denser graphs. For instance, its TPR is close to $0.7$ on graphs with degree of $5$. Similar observations can be made for the case of $n = 10000$. These may be ascribed to the approximate violations of faithfulness in the simulated data, and therefore MMPC may miss some neighbors because of unfaithful conditional independencies. On the other hand, GLasso requires only the SSCF assumption, which is intuitively much weaker than faithfulness.

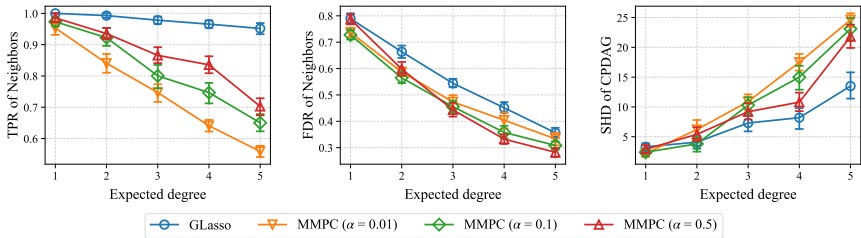

Figure 3: Results of different super-structure estimation methods on 10-node graphs with different degrees. The sample size is $n = 300$. Lower is better, except for TPR.

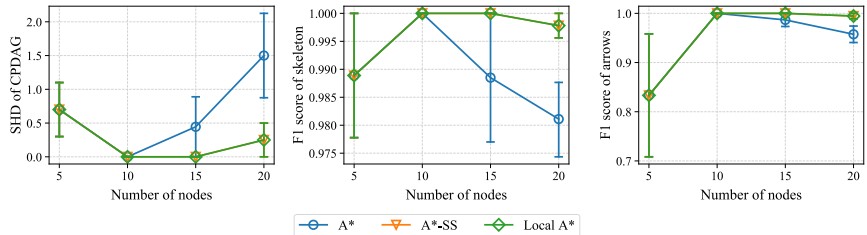

Figure 4: Validation of the proposed methods on graphs with expected degree of 2. The sample size is $n = 10000$. For A*-SS and Local A*, it is assumed here that the ground truth $\mathrm{supp}(\mathbf{\Theta})$ is known. Higher is better, except for SHD.

We also consider the use of the structures estimated by GLasso and MMPC as super-structures for A*-SS, to study how they affect the exact search procedure. The third panels of Figures 3 and 7 show that the super-structures estimated by GLasso lead to better search results than MMPC, especially on graphs with high expected degrees. This is because the performance of recovering the true CPDAGs is upper-bounded by the proportion of direct neighbors discovered by the super-structure.

## 5.2  Validation of the Proposed Search Strategies

We now conduct experiments to study the asymptotic correctness of the proposed strategies, by comparing the A*-SS and Local A* methods to A*. For A*-SS and Local A*, we assume that the support of the true inverse covariance matrix is known to ensure that the exact search does not make errors just because of an inaccurate estimate of the inverse covariance. In Sections 5.3 and 5.4, we consider the more realistic setting in which the ground truth $\mathrm{supp}(\mathbf{\Theta})$ is not known. Here we simulate $n = 10000$ samples for the graphs with expected degree of 2 and $\{5, 10, 15, 20\}$ nodes.

The results are reported in Figure 4, which show that the performance of Local A* and A*-SS is consistent across all metrics, including the F1 score of undirected and directed edges. This indicates that Local A* is able to output the right MECs that are the same as A*-SS in most cases, which suggests that Local A* is asymptotically correct. It is not surprising that these two methods perform better than A* for graphs with 15 and 20 nodes, because we assume here that the true $\mathrm{supp}(\mathbf{\Theta})$ is known, which greatly reduces the search space and therefore is less susceptible to statistical errors owing to finite samples. This implies that a sound super-structure not only improves the efficiency, but also the performance. On the other hand, although A* is guaranteed to find the global optimum, it may still miss or incorrectly estimate some edges because of statistical errors.

## 5.3  Comparison with Other Baselines

To compare the proposed methods, i.e., A*-SS and Local A*, with the baselines, we consider graphs with varying sample sizes $n \in \{300, 10000\}$ and expected degrees from $\{1, 2, 3, 4, 5\}$. The baselines include Triplet A*, MMHC, PC, FGES, and SP. We start with 7-node graphs since the computation of SP may be too slow on graphs larger than that.

Due to space limit, we report the results for 300 and 10000 samples in Figures 8a and 8b, respectively, in Appendix E.4. With $n = 300$ samples, A*-SS and Local A* outperform other baselines across all metrics in most settings. Consistent with the observation in Section 5.2, the results of them are

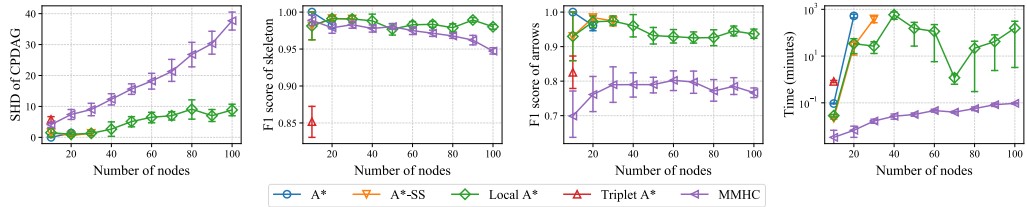

Figure 5: Results and time for graphs with different sizes. The sample size is $n = 10000$. Lower is better, except for F1 score.

exactly the same in all settings, which indicates that, similar to A\*-SS, Local A\* appears to output the globally optimal solutions. With a larger sample size $n = 10000$, the advantage of A\*-SS and Local A\* slightly diminishes as their performance is similar to the other baselines. A possible reason is that approximate violations of faithfulness are less likely to occur with more samples; thus, methods like PC and FGES are able to find the right solutions. This also demonstrates that exact methods like A\*-SS and Local A\* that rely on the SMR assumption are relatively reliable in practice since they are less susceptible to approximate violations of faithfulness. Notice that SP performs the best for $10000$ samples; this is not surprising as it is guaranteed to find the globally optimal solutions. For A\*-SS and Local A\* that rely on super-structure, their performance may be upper-bounded by the proportion of direct neighbors discovered by the super-structure estimation method (see Section 5.1). However, SP does not scale well to large graphs and can handle at most $8$ nodes due to the huge search space.

### 5.4    Scaling up A\* to Large Graphs

We have proposed several strategies to scale up exact methods like A\*, such as local search and constraining the parent graphs based on super-structure. To study the contribution of each strategy, we compare different variants of A\*, including A\*, A\*-SS, and Local A\*. Here we adopt Triplet A\* and MMHC as our baselines since they have been shown to have similar performance to the other baselines in Section 5.3. Furthermore, they are the most relevant to our proposed Local A\* method as they are also two-stage hybrid methods, i.e., they first estimate a super-structure (or skeleton), and then perform score-based search on it. We conduct experiments with different graph sizes up to $300$ nodes, and terminate the experiments that run for more than four days.

For better readability, the results and time for graphs with 100 nodes or less are depicted in Figure 5, and those for larger graphs are provided in Appendix E.5. One observes that Local A\* and MMHC significantly outperform the others in terms of scalability. Both of them can scale up to 300 nodes. Note that A\*-SS scales up to 30 nodes, while A\* can only be scaled up to 20 nodes in our experiments. This verifies that incorporating super-structure with A\* indeed helps reduces search space. Moreover, the search time for Local A\* increases gently in the number of nodes, which demonstrates the potential of its scalability for large graphs. Note that in some of the experiments, Local A\* did not finish within four days and were terminated. As described in Section 4.3, we observe that its running time depends on the maximum size of local clusters. If some of the local clusters contain many variables, the running time may be much longer. On the other hand, MMHC has a shorter running time than Local A\*, because it adopts a hill-climbing strategy and is known to run relatively fast, similar to PC and FGES that generally finish within a few minutes in our experiments. Nevertheless, on large graphs, Local A\* has much better structure recovery results than MMHC.

## 6    Conclusion

We studied the problem of reliable causal discovery with assumptions weaker than faithfulness. Specifically, in our proposed procedure, we adopted exact search that requires the SMR assumption, and relaxed the faithfulness assumption as much as is viable. We developed (1) a sound super-structure estimation method based on the SSCF assumption that is intuitively much weaker than faithfulness, (2) an improved exact score-based method with the constraint of the estimated super-structure, and (3) a local search strategy that improves the scalability of exact search. The efficacy of the proposed method has been validated in our experiments conducted across various settings. It is worth noting that our procedure, at least in its current form, works only in the linear Gaussian setting. Therefore, a future direction is to extend it to the non-Gaussian and discrete cases.

## Acknowledgments

The authors would like to thank the anonymous reviewers for helpful comments and suggestions. This work was supported in part by the National Institutes of Health (NIH) under Contract R01HL159805, by the NSF-Convergence Accelerator Track-D award #2134901, by the United States Air Force under Contract No. FA8650-17-C7715, and by a grant from Apple. The NIH or NSF is not responsible for the views reported in this article. JZ's research was supported in part by the RGC of Hong Kong under GRF 13602720.

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
