# Appendices

## A  Further Common Assumptions for Causal Discovery

Following from Section 2.1, we review several relaxations of faithfulness, which give rise to different constraint-based causal discovery methods that allow certain type of unfaithfulness.

**Assumption 8 (Adjacency-faithfulness [30]).** *Given a DAG $\mathcal{G}$ and distribution $\mathbb{P}$ over the variable set $\mathbf{V}$, if two variables $X$ and $Y$ are adjacent in $\mathcal{G}$, then they are dependent conditional on any subset of $\mathbf{V} \setminus \{X, Z\}$.*

**Assumption 9 (Orientation-faithfulness [30]).** *Given a DAG $\mathcal{G}$ and distribution $\mathbb{P}$ over the variable set $\mathbf{V}$, let $<X, Y, Z>$ be any unshielded triple in $\mathcal{G}$.*

*(1) If $X \rightarrow Y \leftarrow Z$, then $X$ and $Z$ are dependent conditional on any subset of $\mathbf{V} \setminus \{X, Z\}$ that contains $Y$;*

*(2) otherwise, $X$ and $Z$ are dependent conditional on any subset of $\mathbf{V} \setminus \{X, Z\}$ that does not contain $Y$.*

Under the Markov and adjacency-faithfulness assumptions, any violation of orientation-faithfulness is *detectable* in the sense that the true distribution $\mathbb{P}$ is not faithful to any DAG [30, 49]. This presents a concrete method to detect unfaithfulness, leading to a variation of the PC method known as Conversative PC [30] that avoids making definite claim of the causal structure when violation of orientation-faithfulness is detected. Along this line, a further restriction of the adjacency-faithfulness assumption has been formulated and also adopted by the Very Conservative SGS algorithm [39] to relax the type of faithfulness assumption required.

**Assumption 10 (Triangle-faithfulness [49]).** *Given a DAG $\mathcal{G}$ and distribution $\mathbb{P}$ over the variable set $\mathbf{V}$, let $<X, Y, Z>$ be any three variables that form a triangle in $\mathcal{G}$ (i.e., they are adjacent to one another).*

*(1) If $Y$ is a non-collider on the path $<X, Y, Z>$, then $X$ and $Z$ are dependent conditional on any subset of $\mathbf{V} \setminus \{X, Z\}$ that does not contain $Y$.*

*(2) If $Y$ is a collider on the path $<X, Y, Z>$, then $X$ and $Z$ are dependent conditional on any subset of $\mathbf{V} \setminus \{X, Z\}$ that contains $Y$.*

## B  Analysis of the SUCF Assumption

We provide further simulation results for the analysis of the SUCF assumption in Section 3.2.

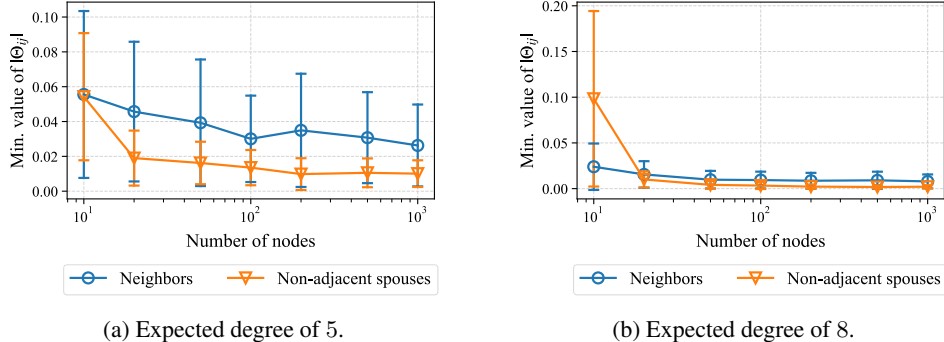

(a) Expected degree of 5.  (b) Expected degree of 8.

Figure 6: Visualizations of the minimum values $\min_{i,j}\{|\boldsymbol{\Theta}_{ij}| : X_i$ and $X_j$ correspond to a pair of neighbors in $\mathcal{G}\}$ and $\min_{i,j}\{|\boldsymbol{\Theta}_{ij}| : X_i$ and $X_j$ correspond to a pair of non-adjacent spouses in $\mathcal{G}\}$ computed across 100 simulations. Different number of nodes and expected degrees are considered. X-axes are visualized in log scale.

# C Proofs

## C.1 Proof of Theorems 1 and 2

We first describe Lemmas 1 and 2 required to prove the main theorems, and these lemmas are similar to Lemma 1 and Theorem 2 by Loh and Bühlmann [20], respectively, but we do assume that $\mathbf{B}$ is strictly upper/lower triangular. Here, Lemma 1 is straightforward from expanding the equation $\Theta = (\mathbf{I} - \mathbf{B})\Omega^{-1}(\mathbf{I} - \mathbf{B})^{\mathsf{T}}$ with $\Omega = \operatorname{diag}(\sigma_1^2, \ldots, \sigma_d^2)$.

**Lemma 1.** *Given a DAG $\mathcal{G}$ and distribution $\mathbb{P}$ that follow a linear Gaussian model with inverse covariance matrix $\Theta$. The entries of $\Theta$ are given by*

$$\Theta_{jk} = -\sigma_j^{-2}\mathbf{B}_{kj} - \sigma_k^{-2}\mathbf{B}_{jk} + \sum_{\ell \neq j,k} \sigma_\ell^{-2}\mathbf{B}_{j\ell}\mathbf{B}_{k\ell}, \qquad \forall j \neq k,$$

$$\Theta_{jj} = \sigma_j^{-2} + \sum_{\ell \neq j} \sigma_\ell^{-2}\mathbf{B}_{j\ell}^2, \qquad\qquad\qquad \forall j.$$

**Lemma 2.** *Given a DAG $\mathcal{G}$ and distribution $\mathbb{P}$ that follow a linear Gaussian model with inverse covariance matrix $\Theta$, the structure defined by $\operatorname{supp}(\Theta)$ is a subgraph of the moralized graph of the true DAG $\mathcal{G}$.*

*Proof.* Let $X_j$ and $X_k$, $j \neq k$ be two variables that are not adjacent in the moralized graph of $\mathcal{G}$. Then it suffices to show that $\Theta_{jk} = \Theta_{kj} = 0$. Clearly, $X_j$ and $X_k$ must not be adjacent in the DAG $\mathcal{G}$, indicating that $\mathbf{B}_{kj} = \mathbf{B}_{jk} = 0$. They also cannot share a common child; otherwise they must be adjacent in the moralized graph. Therefore, we have $\mathbf{B}_{j\ell} = 0$ and $\mathbf{B}_{k\ell} = 0$ for $\ell \neq j, k$. Applying Lemma 1 gives $\Theta_{jk} = \Theta_{kj} = 0$. □

With the above lemmas, we now provide the proofs of the main results.

**Theorem 1.** *Given a DAG $\mathcal{G}$ and distribution $\mathbb{P}$ that follow a linear Gaussian model with inverse covariance matrix $\Theta$, under Markov assumption, the SSCF and SUCF assumptions are satisfied if and only if the structure defined by $\operatorname{supp}(\Theta)$ is the same as the moralized graph of the true DAG $\mathcal{G}$.*

*Proof.* We proceed by contraposition in both parts of the proof.

If part:
Suppose either the SSCF or SUCF assumption is violated, i.e., there exists a collider $X_j \to X_i \leftarrow X_k$ in the DAG $\mathcal{G}$ such that $X_j \perp\!\!\!\perp X_k | \mathbf{V} \setminus \{X_j, X_k\}$. This indicates that $\Theta_{jk} = \Theta_{kj} = 0$. Since $X_j$ and $X_k$ are either neighbors or spouses, there exists an edge between them in the moralized graph of $\mathcal{G}$, but is not contained in the structure defined by $\operatorname{supp}(\Theta)$, showing that they are not the same.

Only if part:
Suppose that the structure defined by $\operatorname{supp}(\Theta)$ is not the same as the moralized graph of $\mathcal{G}$. Then, by Lemma 2, there exists a pair of variables $X_j$ and $X_k$, $j \neq k$ that are adjacent in the moralized graph but $\Theta_{jk} = \Theta_{kj} = 0$. In the linear Gaussian case, we have $X_j \perp\!\!\!\perp X_k | \mathbf{V} \setminus \{X_j, X_k\}$. It remains to consider the following cases:

- If variables $X_j$ and $X_k$ correspond to a pair of non-adjacent spouses in $\mathcal{G}$, then they have an unshielded collider, indicating that the SUCF assumption is violated.

- Otherwise, variables $X_j$ and $X_k$ correspond to a pair of neighbors in $\mathcal{G}$. Assume without loss of generality that $X_j$ is a parent of $X_k$, i.e., $X_j \to X_k \in \mathbf{E}$. This implies that $\mathbf{B}_{jk} \neq 0$ and $\mathbf{B}_{kj} = 0$, which, by Lemma 1, yields

$$\Theta_{jk} = -\sigma_k^{-2}\mathbf{B}_{jk} + \sum_{\ell \neq j,k} \sigma_\ell^{-2}\mathbf{B}_{j\ell}\mathbf{B}_{k\ell} = 0.$$

  Since $\sigma_k^{-2}\mathbf{B}_{jk} \neq 0$, there exists a variable $X_i$ with $i \neq j, k$ such that $\sigma_i^{-2}\mathbf{B}_{ji}\mathbf{B}_{ki} \neq 0$. We then have $\mathbf{B}_{ji} \neq 0$ and $\mathbf{B}_{ki} \neq 0$, indicating that $X_i$ is a common child of the variables $X_j$ and $X_k$. In this case, $X_j \to X_i \leftarrow X_k$ forms a shielded collider in $\mathcal{G}$, which, with the CI relation $X_j \perp\!\!\!\perp X_k | \mathbf{V} \setminus \{X_j, X_k\}$, implies that the SSCF assumption is violated.

In both cases, either the SUCF or SSCF assumption is violated. □

**Theorem 2.** *Given a DAG $\mathcal{G}$ and distribution $\mathbb{P}$ that follow a linear Gaussian model with inverse covariance matrix $\boldsymbol{\Theta}$, under Markov assumption, the SSCF assumption is satisfied if and only if the structure defined by $\mathrm{supp}(\boldsymbol{\Theta})$ is a super-structure of the true DAG $\mathcal{G}$.*

*Proof.* We proceed by contraposition in both parts of the proof. Note that the proof is similar to that of Theorem 1.

If part:
Suppose the SSCF assumption is violated, i.e., there exists a shielded collider $X_j \rightarrow X_i \leftarrow X_k$ in the DAG $\mathcal{G}$ such that $X_j \perp\!\!\!\perp X_k | \mathbf{V} \setminus \{X_j, X_k\}$. This indicates that $\boldsymbol{\Theta}_{jk} = \boldsymbol{\Theta}_{kj} = 0$. Since $X_j$ and $X_k$ are neighbors, there exists an edge between them in $\mathcal{G}$, but is not contained in the structure defined by $\mathrm{supp}(\boldsymbol{\Theta})$, showing that it is not a super-structure of the true DAG $\mathcal{G}$.

Only if part:
Suppose that the structure defined by $\mathrm{supp}(\boldsymbol{\Theta})$ is not a super-structure of the DAG $\mathcal{G}$. Then, there exists a pair of neighbors $X_j$ and $X_k$, $j \neq k$ in $\mathcal{G}$ such that $\boldsymbol{\Theta}_{jk} = \boldsymbol{\Theta}_{kj} = 0$. In the linear Gaussian case, we have $X_j \perp\!\!\!\perp X_k | \mathbf{V} \setminus \{X_j, X_k\}$. Assume without loss of generality that $X_j$ is a parent of $X_k$, i.e., $X_j \rightarrow X_k \in \mathbf{E}$. This implies that $\mathbf{B}_{jk} \neq 0$ and $\mathbf{B}_{kj} = 0$, which, by Lemma 1, yields

$$\boldsymbol{\Theta}_{jk} = -\sigma_k^{-2}\mathbf{B}_{jk} + \sum_{\ell \neq j,k} \sigma_\ell^{-2}\mathbf{B}_{j\ell}\mathbf{B}_{k\ell} = 0.$$

Since $\sigma_k^{-2}\mathbf{B}_{jk} \neq 0$, there exists a variable $X_i$, $i \neq j, k$ such that $\sigma_i^{-2}\mathbf{B}_{ji}\mathbf{B}_{ki} \neq 0$. We then have $\mathbf{B}_{ji} \neq 0$ and $\mathbf{B}_{ki} \neq 0$, indicating that $X_i$ is a common child of the variables $X_j$ and $X_k$. In this case, $X_j \rightarrow X_i \leftarrow X_k$ forms a shielded collider in $\mathcal{G}$, which, with the CI relation $X_j \perp\!\!\!\perp X_k | \mathbf{V} \setminus \{X_j, X_k\}$, implies that the SSCF assumption is violated. $\square$

## C.2 Proof of Theorem 3

**Theorem 3.** *Exact score-based search with BIC asymptotically outputs a DAG that belongs to the MEC of the true DAG $\mathcal{G}$ if and only if the DAG $\mathcal{G}$ and distribution $\mathbb{P}$ satisfy the SMR assumption.*

*Proof.* We provide a proof by contrapositive in both directions based on the consistency of the BIC score [12, 3].

If part:
Suppose that exact score-based search asymptotically outputs a DAG $\mathcal{H}$ (having the highest BIC score) that does not belong to the MEC of the true DAG $\mathcal{G}$. Since the BIC score is known to be consistent, $(\mathcal{H}, \mathbb{P})$ must satisfy the Markov assumption, because otherwise its BIC score is lower than that of the true DAG $\mathcal{G}$ and exact search would not have output $\mathcal{H}$. By assumption, the BIC score of $\mathcal{H}$ is higher than that of $\mathcal{G}$, which, by the consistency of BIC, implies that $|\mathcal{H}| \leq |\mathcal{G}|$, and therefore, $(\mathcal{G}, \mathbb{P})$ does not satisfy the SMR assumption.

Only if part:
Suppose that $(\mathcal{G}, \mathbb{P})$ does not satisfy the SMR assumption. Then there exists a DAG $\mathcal{H}$ not in the MEC of $\mathcal{G}$ such that $|\mathcal{H}| \leq |\mathcal{G}|$, and $(\mathcal{H}, \mathbb{P})$ satisfies the Markov assumption. Without loss of generality, we choose $\mathcal{H}$ with the least number of edges. We first consider the case in which $|\mathcal{H}| < |\mathcal{G}|$. Since both $\mathcal{H}$ and $\mathcal{G}$ satisfy the Markov assumption, by the consistency of BIC, the BIC score of $\mathcal{H}$ is higher than that of $\mathcal{G}$, which implies that exact score-based search will not output any DAG from the MEC of $\mathcal{G}$. For the case with $|\mathcal{H}| = |\mathcal{G}|$, since they are both Markov with distribution $\mathbb{P}$, they have the same BIC score. Therefore, exact search will output a DAG that belongs to the MEC of either $\mathcal{H}$ or $\mathcal{G}$, and is not guaranteed to output a DAG from the MEC of the true DAG $\mathcal{G}$. $\square$

# D Implementation Details

This section provides the implementation details of the proposed Local A* method and the baselines.

## D.1 Local A*

Local A* first uses inverse covariance estimation to discover a super-structure of the underlying DAG, and then applies the A*-SS method on the local clusters formed by each variable and its neighbors within two hops in the super-structure, using some further strategy to reduce search

space (see Section 4.3). In our experiments, we use GLasso to estimate the support of the inverse covariance matrix, implemented through the `scikit-learn` package [26]. We set the coefficient of $\ell_1$ penalty term to $0.05$ for 20-node graphs or smaller, and otherwise to $0.2$. For graphs with more than 40 nodes, a thresholding step is applied on the estimated covariance matrix to remove the entries whose absolute values are less than $0.03$. We use relatively small values for conservative variable selection. If needed, one may further use suitable model selection methods (e.g., cross-validation) to select these hyperparameters. Since we focus on the linear Gaussian setting, we use the BIC score for the exact search procedure. To accelerate Algoritnm 1, we run the local search procedure on 12 local clusters in parallel (i.e., on 12 CPUs). The code is available at `https://github.com/ignavierng/local-astar`.

## D.2  Baselines

The implementation details of the baselines are described below:

- PC and FGES are implemented using the `py-causal` package [33] distributed under the LGPL 2.1 license. For the former, we use the Conservative PC algorithm [30] with Fisher Z test, while the BIC score [34] is adopted for the latter.

- The implementation of MMHC is available through the `bnlearn` package [35] in R that is published under the GPL-3 license.

- SP is implemented using the `causaldag` package in Python under the 3-Clause BSD License.

- We use our own Python implementation of Triplet A* as we were not able to run the official C++ implementation on graphs with more than 5 nodes. Similar to the proposed Local A* approach, parallel computing is used to accelerate the iterative search procedure of Triplet A*.

We use the default hyperparameters unless otherwise stated. For a fair comparison, we run all experiments of the baselines on 12 CPUs.

# E  Supplementary Experiment Results

## E.1  Exact Violations of Faithfulness

To demonstrate the efficacy of GLasso for estimating super-structures with weaker assumptions, we experiment with several examples by using GLasso and MMPC to discover the direct neighbors of the true DAG when faithfulness is guaranteed to be violated. Consider an example where $X \to Y \to Z \to W$ and $X \to W$ such that $X \perp\!\!\!\perp W$ due to path cancellation. In this case, the faithfulness assumption is violated but SSCF holds. In particular, we consider the linear SEM $X = N_X, Y = X + N_Y, Z = Y + N_Z, W = -X + Z + N_W$ where $N_X, N_Y, N_Z, N_W \sim \mathcal{N}(0,1)$. We conduct 100 random simulations for 20, 100, and $10^6$ samples. For 20 samples, GLasso is able to discover all the direct neighbors (in the true DAG) in 98 of the simulations, whereas for 100 and $10^6$ samples, it discovers the neighbors in all simulations. For MMPC, it fails to discover the edge between the direct neighbors $X$ and $W$ in 69, 56, and 98 of the simulations for 20, 100, and $10^6$ samples, respectively, because of the unfaithful independency $X \perp\!\!\!\perp W$. This demonstrates that GLasso is able to recover the direct neighbors in cases where MMPC fails, as the former requires only the SSCF assumption that is intuitively much weaker than faithfulness required by the latter.

## E.2  Structural Intervention Distance

We use SHD and F1 score to compare the different CPDAGs. If we are given DAGs, then we can compute their corresponding SID in a straightforward way. Comparing CPDAGs with SID is more complicated, although it is doable according to Peters and Bühlmann [28, Section 2.4.2], since it requires computing the lower and upper bounds of the SID. For this reason, we did not include the results of SID in this paper. Nevertheless, the observations of SID are nearly identical to those based on SHD. For instance, for the dataset considered in Figure 8a (i.e., 7-node graphs with 300 samples) with expected degree of 2, the average lower and upper bounds of the SID for A*-SS, Local A*, Triplet A*, PC, FGES, MMHC, and SP are $(7.5, 10.5)$, $(7.5, 10.5)$, $(10.2, 13.1)$, $(15.0, 20.1)$, $(14.2, 17.8)$, $(16.2, 18.4)$, and $(14.3, 18.4)$, respectively.

### E.3 Different Super-Structure Estimation Methods

This section provides further empirical results to compare the efficacy of GLasso and MMPC for estimating super-structures in practice (see Section 5.1).

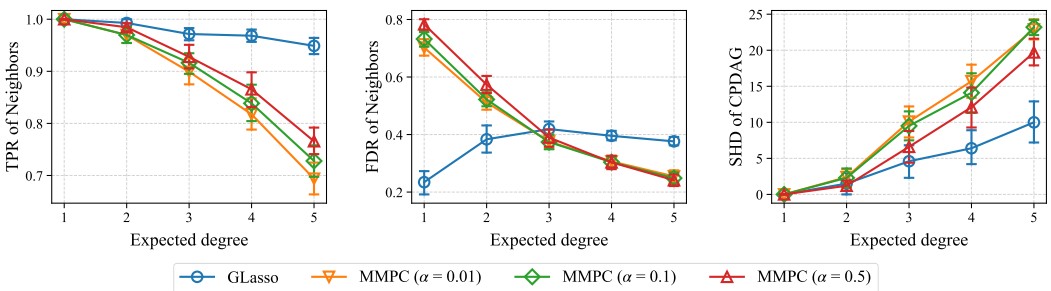

Figure 7: Results of different super-structure estimation methods on 10-node graphs with different degrees. The sample size is $n = 10000$. Lower is better, except for TPR.

### E.4 Comparison with Other Baselines

This section provides further empirical results to compare the proposed methods to the baselines (see Section 5.3).

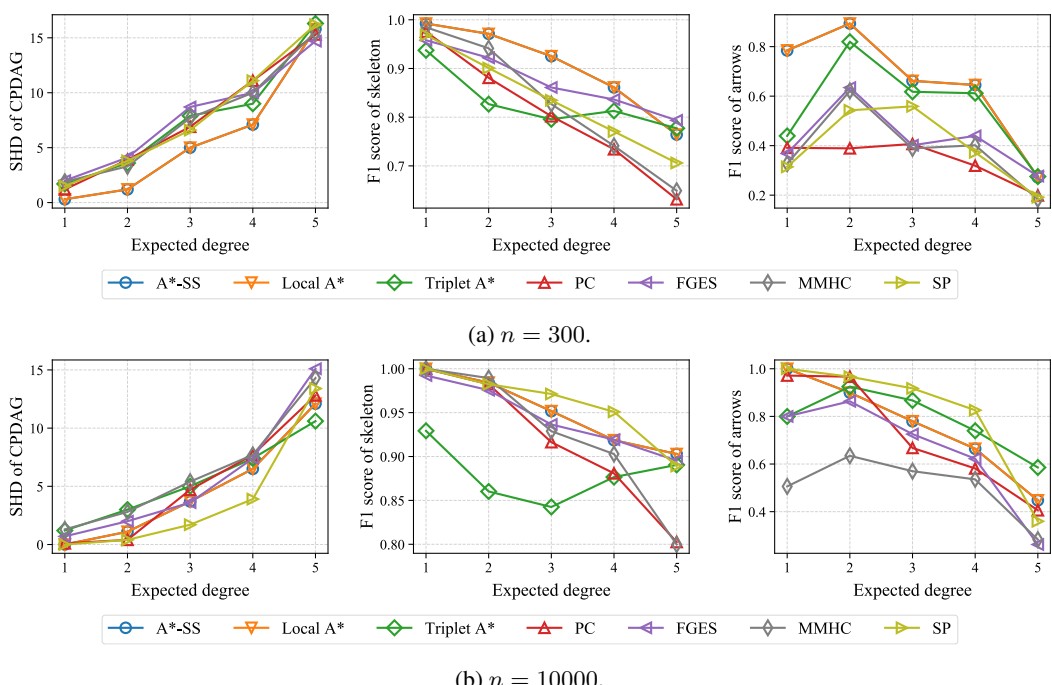

Figure 8: Results of different structure learning methods on 7-node graphs with different degrees and sample sizes. Higher is better, except for SHD. For better visualization, we do not include the standard errors here as each panel has a number of lines.

## E.5  Scaling up A* to Large Graphs

This section provides empirical results for Local A* and MMHC on graphs with $\{50, 100, 150, 200, 250, 300\}$ nodes (see Section 5.4).

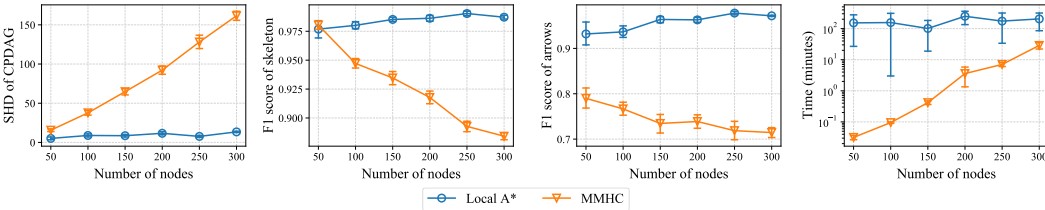

Figure 9: Results and time for Local A* and MMHC on graphs with different sizes. The sample size is $n = 10000$. Lower is better, except for F1 score.