# OpenReview forum: "Reliable Causal Discovery with Improved Exact Search and Weaker Assumptions"
_NeurIPS.cc/2021/Conference — NeurIPS 2021 Poster_

### Official Review · Reviewer_hS1q · 2021-06-30

**Rating:** 7
**Confidence:** 4

**Summary:**

This paper improves the scalability of exact methods for structural learning of Bayesian networks.
The author/s restrict/s the search space via learning a sound super-structure, and then by performing local exact search on the individual variables with their neighbors within two hops in the super-structure.
The paper theoretically proves that the developed approach guarantees correct results under the assumption of infinite data when weaker assumptions than faithfulness are considered.
The last part of the paper describes results of a rich set on numerical experiments, synthetic experiments, which support the effectiveness of the proposed approach which scales up to hundreds of nodes while achieving high accuracy.

**Limitations And Societal Impact:**

I was not able to find any discussion about limitations of the proposed approach.
Some mention is made about limitations of the proposed approach but I think that the author/s should try to better frame and criticize the limitations of the developed approach and which is the expected impact of their work, which I think is relevant.
In particular, I read discussion about numerical results, but still I think something more specific to be concluded when evaluating judging the proposed apporoach.
The paper devotes some space to potential extensions and future works but no mentions to critics of the proposed approach.
The impact to the reseach community could be good but this strongly depends on the fact that the proposed apporach works for a limited class of models.


**Main Review:**

Originality:
The paper is original because it proposes to weaken the standard assumption of faithfulness which is typically the foundation to ensure correcteness when recovering the structure of bayesian networks.
Furtherore, the paper also develops a new algorithm i.e.,a local A* approach for causal discovery.

Quality:
The paper is well structured and written. Terms are properly introduced and defined before using them.
The paper reads very well and the narration is sensible while each piece of information is appropriately presented and discussed.
Technical quality is very good in my humble opinion, the paper shows tha the author/s know very well the research area and also very recent results from the specialized literature.
I did not check the proof of theorems but according to theorems in the paper, the author/s convinced me that their algorithm works.

Clarity:
The paper is clear, in other terms it is is clear the theoretical problem which is tackled and it is also very clear how the problem has been solved.
I also appreciated the numerical experiments section which shows on synthetic data how the proposed approach works and performs.

Significance:
The paper is significant to the community of machine learning with specific reference to bayesian networks and causal networks structural learning from data.
However, the contibution is limited to a very specifc class of models and thus its' potential impact should be retrospectively evaluated.

**Time Spent Reviewing:**

9

---

> ### Author Response · Authors · 2021-08-09
> **We are grateful for the reviewer's effort and the positive comment on our paper.**
>
> We are grateful for the reviewer's effort and the positive comment on our paper. Please find the response to your questions below.
>
> Q1: More structured discussion about limitations and critics of the proposed approach.
>
> A1: We are very grateful for the comment and suggestion. As the reviewer mentioned, the main limitation of the proposed approach, at least in its current form, is that it works only for a specific class of DAG model, i.e., linear Gaussian model, although we believe that it is possible to extend the approach to the non-Gaussian and discrete cases in the future (see our response to Q3 for Reviewer H5q9). Furthermore, unlike the greedy search procedure, the running time of the exact search procedure is longer, as illustrated in Section 5.3 (L388-390). We have updated the manuscript to include a summary of the limitations, in light of your suggestion.

---

> > ### Comment · Reviewer_hS1q · 2021-08-09
> > **Thank for considering my comments**
> >
> > A truly good paper

---

> > > ### Author Response · Authors · 2021-08-10
> > > **We appreciate your immediate and encouraging feedback**
> > >
> > > Thank you very much.

---

### Official Review · Reviewer_H5q9 · 2021-07-08

**Rating:** 6
**Confidence:** 4

**Summary:**

This paper focuses on finding weaker and realistic identifiability conditions for Gaussian linear structural equation models. More precisely, it provides two weaker single shielded-collider-faithfulness and single unshielded-collider-faithfulness assumptions than the widely-used faithfulness, adjacency-faithfulness, orientation faithfulness and triangle-faithfulness assumptions. Based on the proposed assumptions, it further develops a computationally improved exact score-based search methods, and thus, it can recover a large-scale DAG. Lastly, it compares the proposed method and the state-of-the-art A*, MMHC, PC, FGES, and SP algorithm whose accuracy depends on faithfulness or related conditions.

**Ethical Concerns:**

None.

**Limitations And Societal Impact:**

Not related

**Main Review:**

The paper is well-written and well-organized. Especially, I like the introduction and the background sections that summarize the prior works well. The proposed method balances the accuracy and computational cost well. However, the novelty of the paper is restricted on the Gaussian linear SEMs, and some results seem to be obvious when focusing on Gaussian linear SEMs. Furthermore, it seems that the proposed method might be difficult to be generalized for general DAG models. Lastly, it does not provide the computation complexity of the proposed algorithm that is one of the main contributions.

(1) Theorems 1 and 2 seems to be straight forward:

In the Gaussian linear SEM settings, it is well-known that the off-diagonal entries of (j,k) show the conditional independence relationships between (j, k) given the others (V - {j, k}). Together with Markov condition that j and descent of the collider that is not connected to j is conditionally independent, this implies the Theorems 1 and 2 seem to be direct. In addition, as in Lemma 1 of Loh, Po-Ling, and Peter Bühlmann, 2014, also in the supplementary material, the exact value of the element of the inverse covariance matrix can be expressed in terms of edge weights and error variances.
Omega_{jk} = - \beta_{jk} \sigma_k^2 + \sum_{ l such that (j,k,l) is unshilded collider } \beta_{jl} \beta_{kl} / \sigma_{l}^2.
This directly implies the Theorems 1 and 2.

(2) In general linear SEM settings, a conditional dependence does not imply the non-zero partial correlation. Hence, the proposed method seems to be difficult to be generalized to other linear SEMs.

(3) I agree that SUCF assumption can be restrictive in some finite sample settings, as shown in numerical experiments on page 5 in Figure 2. However, the provided result would be meaningful for Gaussian linear SEMs with special range of \beta_{j,k} and sigma_j. In other words, for some settings of \beta, \sigma (e.g., \beta_{jk} is negative), we can also see that the minimum value of Omega_{jk} for spouses are the bigger than neighbors. In addition, we can also easily find the settings for which SUCF are well-satisfied as the degree increases. Hence, I think this paper should carefully provides the focused settings, and emphasizes why these setting should be considered.

(4) I like Theorem 4 that, in the results produced by exact score-based search on the variable set containing the variable X and its parents, children, spouses, and grandparents, the discovered undirected edges between X and its direct neighbors and the v-structures involving X are asymptotically correct.

However, it seems to be unclear that how to apply the Theorem 4 to the proposed algorithm (Local A*), because the local cluster C_i includes not only parents, children, spouses, and grandparents, but also (i) grandchildren, (ii) spouses of parents that are not it parents, (iii) children of its parents, (iv) spouses of children, and (v) spouses of its spouses.

In addition, it is unclear that the meaning of the ordering \pi in Algorithm 1.

(5) I think the provided numerical experiments in Section 5 do not effectively support one of the main ideas that SSCF-based local A* algorithm performs better than the compared faithfulness-based DAG learning algorithms. It is mainly because the low performance of a method possibly comes from an inappropriate conditional independence test and a  strength of association (dependency) function, score function, greedy search method, or violation of the incoherence assumption.

Hence, it would be better to exploit two types of random Gaussian linear SEMs: (i) violating faithfulness (or related faithfulness) whereas satisfying SSCF in population; and (ii) random Gaussian linear SEMs satisfying faithfulness. And then, compares the performance of the considered algorithms in both limited sample and population settings. Otherwise, it is hard to accept that a better-accuracy of the proposed algorithm comes from weaker identifiability condition. Furthermore, it would be better to provide what conditional independence test and scores for MMPC and FGES, respectively. In addition, it would be also better to explain how to choose appropriate regularization parameter for GLasso.

In addition, it is hard to understand the purpose the numerical experiments in Section 5.1. As I understood, it shows that Glasso is clearly better than MMPC in terms of recovering the super-structure defined by the support of estimated inverse covariance matrix. However, in principle, Glasso recovers the skeleton of the moralized graph, whereas MMPC recover the skeleton of a true graph. Hence, the direct comparison seems to be awkward. Furthermore, MMPC does not require the incoherence condition that is almost necessary condition for GLasso.

Lastly, it seems to be unfair that greedy-search- or forward-selection-based PC, FGES, MMHC were compared in terms of accuracy. However, the PC and FGES were not considered when comparing the run-time. As you also know, they are designed for large-scale graphs with thousands node.

Minor:

Definitions of neighbors are different on page 5 where it is the set of nodes including parents and children and in Figure 3 where it seems to be a set nodes including parents, children, and spouses.

Line 370, it would be sample version of the faithfulness assumption, so called \alpha-faithfulness.

########## Updated ###########
Thanks for the kind answer. Based on your asnwer and other reviews, I increase my rate fomr 4 to 6. This paper has a clear contribution, but I thought it is not enough for the NIPS level. However, after reading the other reviews, I changed the my opinion.

By the way, the references are not requried for the definition of the neighbors, but its definition should be clarified in the paper.

In addition, there is no statistical guarantees that GLasso performs better than MMPC in terms of recovering direct neighbors. In fact, MMPC is consistent, whereas GLasso is not when recovering neighbors. In addition, there are many settings in which MMPC empirically performs better than GLasso. However, I agree that the simulation settings used in the paper are widely exploited in the relevant sutides. This is my simple curiosity that why most related studies apply these settings.

**Time Spent Reviewing:**

12 - 24 hours

---

> ### Author Response · Authors · 2021-08-09
> **We greatly appreciate the reviewer's thorough and constructive comments.**
>
> We greatly appreciate the reviewer's thorough and constructive comments, many of which will help improve the readability of our paper. We attempt to address all the concerns in the following.
>
> Q1: Computational complexity of the proposed algorithm.
>
> A1: Thank you for asking this question. Our approach can be easily parallelized by running the local search for all variables in parallel. For instance, with enough CPUs, the running time of the proposed method mainly depends on the maximum size of local clusters. At the same time, it is very challenging to derive the exact computational complexity of the proposed method, as is the case for the A* algorithm, partly owing to the heuristic involved (Section 4, Hart, Nilsson, and Raphael, 1968). This is because the computational complexity is affected by different factors such as the size of sparse parent graphs and the heuristic function (i.e., the dynamic k-cycle conflict heuristics) (Yuan and Malone, 2013). Therefore, we are only able to provide the running time as a proxy of the computational complexity (Figures 5 and 9). We will include a discussion to make this explicit in Section 4.2.
>
> Q2: "Theorems 1 and 2 seem to be straightforward"
>
> A2: Thanks for sharing a nice intuition behind Theorems 1 and 2. We agree that Theorems 1 and 2 are not a surprise. At the same time, they provide a formal way to support the proposed approach in the paper, which is necessary. We first gave the result by Loh and Bühlmann (2014) in Lemma 1 (L575-578), from which Theorem 1 follows. We then went further to formulate precisely the required assumptions from a causal discovery perspective and relate them to different types of colliders. This sheds light on how weak they are as compared to faithfulness. As a consequence, we are able to further weaken the assumptions required for Theorem 1 and obtain Theorem 2 based on only the SSCF assumption to recover a super-structure of the true DAG, which we believe should be provided as a separate result to support our approach.
>
> By the way, in your comment "Omega_{jk} = - \beta_{jk} \sigma_k^2 + \sum_{ l such that (j,k,l) is unshilded collider } \beta_{jl}\beta_{kl} / \sigma_{l}^2", we are wondering whether "unshielded collider" should actually mean "collider" (containing both shielded and unshielded ones). If there is any misunderstanding, please kindly let us know.
>
> Q3: Generalization to other SEMs.
>
> A3: Thanks for pointing out this important issue, which has been considered as future work in L406, because, as you mentioned, it is not straightforward to generalize the proposed method to the general SEMs. Inspired by several related works, there are several possibilities to achieve that. For instance, we plan to extend the method to the linear non-Gaussian case, where some recent works have been proposed to associate the 'generalized precision' with the conditional independencies based on transport maps (Morrison et al., 2017; Baptista et al., 2021). In addition, it is also possible to extend it to the discrete case, where the support of the inverse covariance reflects the conditional independencies under certain assumptions (Loh and Wainwright, 2013).
>
> Q4: Elaboration of the settings in Figure 2.
>
> A4: Thank you for raising this concern. We agree it is possible to find some other settings that "the minimum value of Omega_{jk} for spouses are the bigger than neighbors", particularly when the edge weights are large. We also agree that the observation from Figure 2 is based on a certain range of edge weights and noise variances. At the same time, in the real world, the edge weights and noise variances are not known, and it is possible that the data distribution falls into the setting we considered, leading to violation of the SUCF assumption. Therefore, we believe that Theorem 2 with only the SSCF assumption is more reliable than Theorem 1 in practice as it requires strictly weaker assumptions. We will include a discussion to address this concern in Section 3.2.
>
> Q5: "It seems to be unclear that how to apply the Theorem 4 to the proposed algorithm"
>
> A5: From the statement "the variable set containing the variable X and its parents, children, spouses, and grandparents" (L287-288), we intend to convey that the variable set is a superset of the latter variables. We have updated the statement such that it is clearer how to apply Theorem 4 to the proposed local A* approach.
>
> Q6: "It is unclear that the meaning of the ordering \pi in Algorithm 1"
>
> A6: Thanks for spotting this! It refers to the variable order obtained by sorting the variables based on the size of their local clusters. We will give an explicit definition of this terminology in the revision.
>
> Q7: It would be better to exploit two types of random Gaussian linear SEMs regarding different assumptions.
>
> A7: Thank you for this helpful suggestion. Following your suggestion, we experimented with several examples by using GLasso and MMPC to discover the direct neighbors of the true DAG (please see our response to Q10) when faithfulness is guaranteed to be violated. Consider an example where $X\rightarrow Y\rightarrow Z \rightarrow W$ and $X\rightarrow W$ such that $X\perp W$ due to path cancellation. In this case, the faithfulness assumption is violated but SSCF holds. In particular, we consider the linear SEM $X = N_X, Y = X + N_Y, Z = Y + N_Z, W = - X + Z + N_W$ where $N_X, N_Y, N_Z, N_W \sim \mathcal{N}(0, 1)$. Using this SEM, we conducted $100$ random simulations for different sample sizes, i.e., $20$, $100$, and $10^6$ samples. For $20$ samples, GLasso is able to discover all the direct neighbors (in the true DAG) in $98\%$ of the simulations, whereas for $100$ and $10^6$ samples, it discovers the neighbors in all simulations. For MMPC, it fails to discover the edge between the direct neighbors $X$ and $W$ in $69\%$, $56\%$, and $98\%$ of the simulations for $20$, $100$, and $10^6$ samples, respectively, because of the unfaithful independency $X\perp W$. This supports the empirical study in Section 5.1 (L350-353) that GLasso is better than MMPC in terms of recovering as many direct neighbors as possible, as GLasso requires only the SSCF assumption that is intuitively much weaker than faithfulness required by MMPC. We will include such experiments in Section 5.
>
> Q8: Types of conditional independence test and scores for MMPC and FGES, respectively.
>
> A8: Partial correlation test and BIC score are used for MMPC and FGES, respectively. We will include the details in the revised paper.
>
> Q9: The strategy to choose the regularization parameter for GLasso.
>
> A9: Thanks for the suggestion. As stated in L694-695, we set the parameter to $0.05$ for $20$-node graphs or smaller, and otherwise to $0.2$. We used a relatively small value for conservative variable selection. If needed, one may further use suitable model selection methods (like cross-validation) to select these values. We will make it more explicit in the paper.
>
> Q10: Numerical experiments in Section 5.1.
>
> A10: Thanks for raising the concerns. We are afraid that there might be some misunderstanding--please note that the purpose of the experiments in Section 5.1 is to demonstrate that GLasso is better than MMPC in terms of recovering as many direct neighbors (in the true DAG) as possible (L337-339), but not "recovering the super-structure defined by the support of estimated inverse covariance matrix", which also includes non-adjacent spouses. We conducted this comparison because the final search performance is upper-bounded by the proportion of direct neighbors discovered by the super-structure. Note that the word "neighbors" here refers to direct neighbors (in the true DAG) which include only parents and children, but not the non-adjacent spouses (see our response to Q12).  We have updated Section 5.1 to make the motivation and comparison clear.
>
> Q11: Running time of PC and FGES.
>
> A11: We will include the running time of PC and FGES in the revised version. For $100$-node graphs, PC and FGES finish within $10$ and $1$ minutes, respectively.
>
> Q12: Definition of "neighbors".
>
> A12: The word "neighbors" here refers to direct neighbors (in the true DAG) which include only parents and children, but not the non-adjacent spouses.
>
> References:
> R. Baptista, Y. Marzouk, R. E. Morrison, and O. Zahm. Learning non-Gaussian graphical models via hessian scores and triangular transport. arXiv preprint arXiv:2101.03093, 2021.
>
> P.-L. Loh and P. Bühlman. High-dimensional learning of linear causal networks via inverse covariance estimation. Journal of Machine Learning Research, 2014.
>
> P.-L. Loh and M. J. Wainwright. Structure estimation for discrete graphical models: Generalized covariance matrices and their inverses. The Annals of Statistics, 2013.
>
> R. Morrison, R. Baptista, and Y. Marzouk. Beyond normality: Learning sparse probabilistic graphical models in the non-Gaussian setting. In Advances in Neural Information Processing Systems, 2017.
>
> P. Hart, N. Nilsson, and B. Raphael. A Formal Basis for the Heuristic Determination of Minimum Cost Paths. IEEE Transactions on Systems Science and Cybernetics, 1968.
>
> C. Yuan and B. Malone. Learning optimal Bayesian networks: A shortest path perspective. Journal of Artificial Intelligence Research, 2013.

---

> ### Author Response · Authors · 2021-08-14
> **We are very grateful for your positive and thoughtful feedback**
>
> We are very grateful for your positive and thoughtful feedback. Your comments certainly improved the quality and readability of our paper. We will carefully update the manuscript and add the newly introduced references according to all these comments.
>
> Regarding the definition of the neighbors, we totally agree with you and will clarify it explicitly in the paper. And we would like to appreciate your insightful opinions about MMPC and GLasso, which are worth exploring further.
>
> We once again appreciate your efforts in reviewing this manuscript.

---

### Official Review · Reviewer_BZ1f · 2021-07-12

**Rating:** 7
**Confidence:** 4

**Summary:**

This paper proposes two steps for learning the DAG structure from linear Gaussian data. First, it is shown that under a condition weaker than several forms of faithfulness, the inverse of the observational covariance matrix can be used to find a graph guaranteed to contain all edges in the DAG. Second, another theoretical result is established that shows that an exact structure learning algorithm will find the correct edges and v-structures around X even when it sees only a particular neighbourhood around X. An algorithm is proposed that leverages both results.

**Limitations And Societal Impact:**

Adequately addressed.

**Main Review:**

I think several results in this paper are interesting and novel. In particular, Theorem 4 (guaranteeing partial correctness of local search results) seems to be a powerful new result. I also appreciate how they fit together in this paper. There are a few points where the presentation could be improved, detailed below.

It should be mentioned earlier on (preferably in the abstract) that this paper deals with linear SEMs, not e.g. discrete BNs. The technique of using the inverse of the covariance matrix is very specific to the linear case.

What other methods exist for structure learning that rely on the inverse of the covariance matrix? How does the proposed method compare to them?

Near Figure 2, the term "spouse" is used, and apparently (no definition is given) pairs of adjacent nodes are not considered spouses. I don't think that's the standard definition.

There is something subtly wrong with Theorem 2. Under standard definitions, it is allowed to have B_ij = 0 even though there is an edge there. Such a case is a counterexample to the theorem. It also goes against the SMR assumption, and I think that by assuming SMR in the theorem's statement, the problem can be fixed. Instead of the implication "edge in E implies B is not 0" which is currently used in the proof, the SMR assumption can be invoked at that point to get to the same conclusion.

In the paragraph below Theorem 3, the text about parent graphs isn't very informative: if the reader doesn't know what they are already, they won't learn from reading this.

Spelling / grammar:
34: charactezation - letters missing
55: constraint -> constrain
210: the strong SUCF assumption -> a strong SUCF assumption; consider adding quotes around "strong SUCF assumption"

UPDATE: I appreciate that the authors are incorporating my comments into their manuscript. (My score didn't change.)

**Time Spent Reviewing:**

3

---

> ### Author Response · Authors · 2021-08-09
> **We thank the reviewer for the positive feedback and time devoted to our work.**
>
> We thank the reviewer for the positive feedback and time devoted to our work. Our responses to these comments are provided below.
>
> Q1: "It should be mentioned earlier on that this paper deals with linear SEMs"
>
> A1: Thanks for this suggestion. We have updated the manuscript to make it clear in both the abstract and the introduction. In particular, in the second paragraph of the introduction, we mention that our work focuses on the linear Gaussian case.
>
> Q2: "What other methods exist for structure learning that rely on the inverse of the covariance matrix?"
>
> A2: To the best of our knowledge, the closest work that relies on the inverse covariance matrix is (Loh and Bühlmann, 2014), which we discussed in L183-184. We will further make the key differences between our approach and theirs more explicit in the paper. In particular, they adopt a type of faithfulness assumption to guarantee that the structure defined by the inverse covariance is the same as the moralized graph of true DAG, similar to Theorem 1 in our work. However, we focus on formulating precisely the required assumption from a causal discovery perspective, to shed light on how weak they are as compared to faithfulness. This enables us to further weaken the assumptions required and obtain Theorem 2 based on only the SSCF assumption to recover a super-structure of the true DAG based on inverse covariance. This leads to the proposed local A* approach based on two-hop neighbors in the super-structure with weaker assumptions.
>
> Q3: "Under standard definitions, it is allowed to have B_ij = 0 even though there is an edge there"
>
> A3: Thank you for raising this concern. According to L139-140, we have made the structural minimality assumption (Peters et al., 2017, Remark 6.6) on the relationships between the absence of edges and the zero coefficients. Following your suggestion, we have updated the manuscript to make this explicit in the paper.
>
> Q4: Regarding the definition of ‘spouses’ and the explanation of ‘parent graphs’.
>
> A4: Thank you for pointing these out. We will make these terminologies clear in the revised version.
>
> References:
> P.-L. Loh and P. Bühlman. High-dimensional learning of linear causal networks via inverse covariance estimation. Journal of Machine Learning Research, 2014.
>
> M. Forster, G. Raskutti, R. Stern, et al. The frugal inference of causal relations. The British Journal for the Philosophy of Science, 2018.
>
> J. Peters, D. Janzing, and B. Schölkopf. Elements of Causal Inference - Foundations and Learning Algorithms. MIT Press, 2017.

---

### Official Review · Reviewer_vWU7 · 2021-07-19

**Rating:** 7
**Confidence:** 4

**Summary:**

The authors propose a new method for learning the graph structure of Bayesian networks.  Specifically, the authors propose a method for learning the super-structure and efficient local search methods for producing the final structure given the super-structure.

**Limitations And Societal Impact:**

The discussion here is adequate, given the focus of the paper.

**Main Review:**

In general, the experiments (Section 5) appear well conceived and well executed.  They focus on specific aspects of the overall approach proposed in the paper, and they compare to plausible alternatives.  Specifically, the authors use a reasonable and diverse array of baselines in their experiments.

However, the authors measure the accuracy of structure learning methods using structural Hamming distance (SHD), a measure that has been strongly and appropriately criticized in recent work (Peters & Bühlmann 2015; Gentzel et al. 2019).  The paper would be stronger if it used a more realistic and informative performance measure.  The results are strong enough that the observed differences in SHD are probably indicative of good overall performance, but measures such as structural intervention distance (SID) or measures of the accuracy of actual interventional queries would be more informative.

In the second paragraph of the introduction, the authors state that constraint-based methods “first estimate the skeleton of the causal structure and then identify the v-structures based on conditional independence tests.” Edge orientation rules do more than just identify v-structures, although that is perhaps the most important (and justified) rule for edge orientation.  The description should be updated to be more accurate.

The authors note in the introduction that the limitations of randomized experiments concern ethical issues.  While this is one potential problem with randomized experiments, there are others, including practicality (astronomers cannot randomly assign the size of a galaxy) and cost (it would be prohibitively expensive to randomly assign programmers to multiple teams, each of which develops a major software platforms using a different programming language).

**Time Spent Reviewing:**

1.5

---

> ### Author Response · Authors · 2021-08-09
> **We sincerely thank the reviewer for the time devoted and the helpful comments.**
>
> We sincerely thank the reviewer for the time devoted and the helpful comments. Below we give a point-by-point response to the comments.
>
> Q1: "Measures such as structural intervention distance (SID) or measures of the accuracy of actual interventional queries would be more informative."
>
> A1: Thanks for the helpful comment. Here we use SHD to compare the different CPDAGs. If we are given DAGs, then we can compute their corresponding SID in a straightforward way. Comparing CPDAGs with SID is more complicated, although it is doable according to Peters and Bühlmann (2015, Section 2.4.2), since it requires computing the lower and upper bounds of the SID. For this reason, we did not include the results of SID in the paper. In light of your comments, we have computed the SID for the experiments in Section 5.2, which leads to nearly identical observations to those based on SHD. For instance, for the dataset considered in Figure 4 (i.e., $7$-node graphs with $300$ samples) with expected degree of $2$, the average lower and upper bounds of the SID for Local A*, Triplet A*, PC, FGES, MMHC, and SP are $(7.5, 10.5)$, $(10.2, 13.1)$, $(15.0, 20.1)$, $(14.2, 17.8)$, $(12.2, 16.6)$, and $(14.3, 18.4)$, respectively. We will include a summary of the results in the paper.
>
> Q2: The description of the edge orientation rules and limitations of randomized experiments could be more accurate.
>
> A2: We are very grateful to you for pointing these out. We have updated the manuscript to make these clear in the revised version.
>
> References:
> J. Peters and P. Bühlmann. Structural intervention distance (SID) for evaluating causal graphs. Neural Computation, 2015.

---

### Decision · Program_Chairs · 2021-09-27

**Decision:**

Accept (Poster)

**Comment:**

There is consensus on acceptance among reviewers. There are a few suggestions and discussions that can be helpful to improve the manuscript, in particular to appeal the potential audience at neurips 2021.